# Elucidating the mechanisms of atmospheric new particle formation in the highly polluted Po Valley, Italy

Jing Cai[1], Juha Sulo[1], Yifang Gu[1], Sebastian Holm[1], Runlong Cai[1], Steven Thomas[1], Almuth Neuberger[2], Fredrik Mattsson[2], Marco Paglione[3], Stefano Decesari[3], Matteo Rinaldi[3], Rujing Yin[1], Diego Aliaga[1], Wei Huang[1], Yuanyuan Li[1,4], Yvette Gramlich[2], Giancarlo Ciarelli[1], Lauriane Quéléver[1], Nina Sarnela[1], Katrianne Lehtipalo[1,5], Nora Zannoni[3], Cheng Wu[6], Wei Nie[4], Juha Kangasluoma[1], Claudia Mohr[7,8], Markku Kulmala[1,4,9], Qiaozhi Zha[1,4], Dominik Stolzenburg[1,10*], Federico Bianchi[1*]

[1] Institute for Atmospheric and Earth System Research, Faculty of Science, University of Helsinki, Helsinki 00014, Finland

[2] Department of Environmental Science, Stockholm University, Stockholm 11418, Sweden

[3] Italian National Research Council-Institute of Atmospheric Sciences and Climate (CNR-ISAC), Bologna, 40129, Italy

[4] School of Atmospheric Sciences, Nanjing University, Nanjing, 210023, China

[5] Finnish Meteorological Institute, Helsinki, 00560, Finland

[6] Department of Chemistry and Molecular Biology, Atmospheric Science, University of Gothenburg, Gothenburg 41296, Sweden

[7] Laboratory of Atmospheric Chemistry, Paul Scherrer Institute, Villigen, 5232, Switzerland

[8] Department of Environmental System Science, ETH Zurich, Villigen, 5232, Switzerland

[9] Beijing Advanced Innovation Center for Soft Matter Science and Engineering, Beijing University of Chemical Technology, Beijing 100029, China

[10] Institute for Materials Chemistry, TU Wien, Vienna 1060, Austria

*Correspondence to: federico.bianchi@helsinki.fi and dominik.stolzenburg@tuwien.ac.at*

**Abstract**

New particle formation (NPF) is a major source of aerosol particles and cloud condensation nuclei in the troposphere, playing an important role in both air quality and climate. Frequent NPF events have been observed in heavily polluted urban environments, contributing to the aerosol number concentration by a significant amount. The Po Valley region in northern Italy has been characterized as a hotspot for high aerosol loadings and frequent NPF events in Southern Europe. However, the mechanisms of NPF and growth in this region are not completely understood. In this study, we conducted a continuous 2-month measurement campaign with state-of-the-art instruments to elucidate the NPF and the growth mechanisms in Northern Italy. Our results demonstrate that in this area, frequent NPF events (66% of all days during the measurement campaign) are primarily driven by abundant sulfuric acid ($8.5 \times 10^6$ cm$^{-3}$) and basic molecules. In contrast, oxygenated organic molecules from the atmospheric oxidation of

volatile organic compounds (VOCs) appear to play a minor role in the initial cluster formation but
contribute significantly to the consecutive growth process. Regarding alkaline molecules, amines, are
insufficient to stabilize all sulfuric acid clusters in Po Valley. Ion cluster measurements and kinetic
models suggest that ammonia (10 ppb) must therefore also play a role in the nucleation process.
Generally, the high formation rates of sub-2 nm particles (87 $cm^{-3} s^{-1}$) and nucleation mode growth rates
(5.1 nm $h^{-1}$) together with the relatively low condensational sink ($8.9 \times 10^{-3}$ $s^{-1}$) will result in a high
survival probability of newly formed particles, making NPF crucial for the springtime aerosol number
budget. Our results also indicate that reducing key pollutants such as $SO_2$, amine and $NH_3$, could help
to decrease the particle number concentrations substantially in the Po Valley region.

## 1. Introduction

New particle formation (NPF) occurs ubiquitously in the troposphere and affects the global climate
(Dunne et al., 2016) and local or regional air quality (Kulmala et al., 2021). NPF and further growth of
the newly formed particles dominate aerosol number concentrations and are the major contributor to
the ultrafine (<100 nm) aerosol budget, which poses a significant health threat to the population in
polluted areas (Schraufnagel, 2020). While air pollution mitigation strategies mostly focus on reducing
particulate mass (particulate matter below 2.5 µm ($PM_{2.5}$)), ultrafine particle number concentrations
might not be affected by such policies (De Jesus et al., 2019). It is therefore essential that we understand
the mechanisms leading to NPF in polluted environments to design better targeted air quality strategies
for polluted European regions, where $PM_{2.5}$ reduction measures are already implemented.
NPF is closely linked to atmospheric air pollution. Efficient nucleation and growth are crucial factors
contributing to haze formation, according for over 65% of the particle number concentrations in urban
environment (Kulmala et al., 2021; Guo et al., 2014). Frequent NPF events have also been observed in
heavily polluted urban environments, including megacities in China (Chu et al., 2019; Yao et al., 2018;
Du et al., 2022) and India (Sebastian et al., 2022). Strong and frequent NPF events have been reported
in the most urbanization areas in China, such as the North China Plain (Wang et al., 2015; Wang et al.,
2013; Wu et al., 2011; Wu et al., 2007; Shen et al., 2011), Yangtze River Delta (Dai et al., 2017; Yu et
al., 2016; Xiao et al., 2015) and Peal River Delta (Yue et al., 2013; Peng et al., 2014; Liu et al., 2008).
This observation contradicts theoretical calculations that suggest NPF events are less likely to occur in
polluted areas, where high levels of preexisting aerosols acting as condensational sinks (CS) are capable
of quickly scavenging gaseous precursors of NPF (Kulmala et al., 2017).
The elucidation of NPF precursors and mechanisms has varied among different sampling locations and
studies. No uniform theory or mechanism can elucidate the NPF occurrence in different polluted areas
or in different seasons. For example, in Shanghai and Beijing, China, sulfuric acid (SA, $H_2SO_4$) and
amines were identified as key contributors to initial particle formations (Yao et al., 2018; Cai et al.,
2021; Yan et al., 2021). On the other hand, some studies also suggests that photooxidation products of
vehicle emitted organic vapors, dominate NPF in urban conditions rather than SA or base species (Guo
et al., 2020). Meanwhile, in Barcelona, Spain, which is significantly less polluted than Asian megacities
but still shows frequent high pollution levels, NPF was reported to be associated with SA along with
highly oxygenated organic molecules (HOMs) (Brean et al., 2020). The discrepancies in the reported
NPF mechanisms may arise from the limited utilization of state-of-the-art instruments, such as those
capable of measuring size distribution down to 1-2 nm and directly identifying clusters and vapors with
the influences by spatio-temporal variations (Wang et al., 2017). Therefore, gaining a better knowledge
of the key participants, nucleation mechanism and the roles of pre-existing particles is crucial for
comprehending the causes of the high NPF frequencies in polluted regions. This knowledge can be
essential for developing effective local $PM_{2.5}$ control and implementation strategies.
The Po Valley region is one of the most important industrial and agricultural areas in Southern Europe
with dense population (>17 million/70,000 $km^2$). It is located in northern Italy, surrounded by the Alps
(in the north), the Apennine mountains (in the south), and the Adriatic Sea (in the east). High primary
anthropogenic emissions, a mixture of numerous pollutants from industrial, urban and agricultural
sources, together with frequently occurring stagnant meteorological conditions in winter make the Po
Valley region a hotspot in Europe for high aerosol loadings (Saarikoski et al., 2012; Li et al., 2014;
Finzi and Tebaldi, 1982; Daellenbach et al., 2023). But it is distinct from Asian megacities as the
population density is significantly lower (250 people km$^{-2}$ in Po Valley compared to e.g., 1 400 people
km$^{-2}$ in Beijing), resulting in effects such as traffic or residential heating being less dominant pollution
sources. At the same time, NPF occurs frequently in the Po Valley (Hamed et al., 2007; Manninen et
al., 2010). For example, Shen et al. (2021) observed that NPF events took place on approximately 70%
of the days during spring and summer. Similarly, Kontkanen et al. (2017) discovered that during
summer, NPF occurred on 89% of the days. During NPF event days, high formation rates of sub-2nm
neutral particles ($J_2$, ~$10^1$ to $10^2$ cm$^{-3}$ s$^{-1}$, (Kontkanen et al., 2017)) and SA concentrations (~$1\times10^7$ cm$^{-3}$)
were observed in the Po Valley (Paasonen et al., 2010; Kontkanen et al., 2017). These levels were
among the highest recorded in a study conducted at nine sites across the Northern Hemisphere
(Kontkanen et al., 2017).
While previous studies conducted in the Po Valley have reported frequent NPF events characterized by
high nucleation and growth rates, the clustering mechanism and the dominant precursors for particle
growth have not been investigated to-date. Especially with respect to the distinct features of Po-Valley
compared to the more intensely researched megacity environments, a deeper understanding of frequent
NPF events, including their precursors, nucleation mechanisms, and growth processes is crucial for air
pollution control and the effective implementation of PM$_{2.5}$ mitigation measures in such semi-urban but
highly industrialized regions as Po Valley. In this study, we conducted a 2-month field campaign in the
months of March – April 2022, we 1) identified the chemical composition of atmospheric neutral and
ion clusters by a set of state-of-the-art mass spectrometers, 2) characterized the initial NPF and further
growth rates using particle number size distribution measurement down to 1 nm, and 3) compared the
field measurement results with the recent Cosmics Leaving Outdoor Droplets (CLOUD) chamber
experiments to investigate the mechanism of NPF events in the Po Valley region. This allowed us to
elucidate the NPF and growth mechanisms at a severely polluted Southern European site, and to give
insights in best mitigation strategies for ultrafine particle pollution in the context of already
implemented PM$_{2.5}$ reduction strategies.

## 2. Method

### 2.1 Measurement site

Our measurement was part of the Fog and Aerosol InterRAction Research Italy (FAIRARI) field
campaign in San Pietro Capofiume (SPC, 44.65°N, 11.62°E, 5 m a.s.l.), located in the Po Valley region
in Northern Italy. The measurement site is part of the Aerosol, Clouds and Trace Gases Research
Infrastructure (ACTRIS)-Italy network and operated by the Italian National Research Council-Institute
of Atmospheric Sciences and Climate (CNR-ISAC). The SPC site is approximately 30 km northeast of
Bologna (~400, 000 residents) and 20 km south of Ferrara (~130, 000 residents), the two major cities
in the area. The distance from the measurement site to the Adriatic Sea (to the east) is about 50 km. The
area around the sampling site consists of agricultural fields and a smaller town (<2, 000 inhabitants,
within 5 km) and smaller settlements in the proximity. Given its location, the SPC rural station is
considered to be representative of the regional background of the Po Valley (Paglione et al., 2021;
Paasonen et al., 2010; Hamed et al., 2007; Saarikoski et al., 2012; Decesari et al., 2014; Paglione et al.,
2020). The instruments for the NPF measurement were operated in a temperature controlled (~20 °C)
container from March 1 to April 30, 2022.
During the sampling period, the daily average temperature ranged from 1°C to 17°C. The average wind
speed (WS) was approximately 2.4±1.5 m/s (Fig. 1b). The average WS in the daytime was 3.5 m/s from
the east, which was significantly higher than at night (1.5 m/s) from the west. Strong diurnal variations
of wind direction were observed, which was typically from the west at night and shifted to the east
during the day (Fig. 1a). This pattern was potentially influenced by the sea-land breeze from the Adriatic
Sea. Accordingly, the daily average relative humidity (RH) varied from 41% to 98%, with values as

high as 85% at night, which sharply decreased to around 40% at noon caused by the strong temperature variation.

## 2.2 Instruments

### 2.2.1 Chemical composition measurements

The chemical composition of cluster ions was measured using a high-resolution atmospheric-pressure-interface time-of-flight mass spectrometer (APi-TOF, Aerodyne Research Inc. & Tofwerk AG). The APi-TOF measures naturally charged ions present in the ambient environment. A detailed description of the instrument can be found in Junninen et al. (2010). In this study, ambient air was sampled through a 0.57-meter stainless steel tube with a flow rate of ~10 liters per minute (LPM), with 0.8 LPM of the sample flow entering the APi-TOF.

The concentration of SA was measured using a nitrate ion ($NO_3^-$)-based chemical-ionization (CI) atmospheric-pressure-interface time-of-flight mass spectrometer (CI-APi-TOF, Aerodyne Research Inc. & Tofwerk AG (Jokinen et al., 2012)). The CI-APi-TOF is an APi-TOF coupled with a CI-unit, equipped with a soft X-ray source (L9490, Hamamatsu's 9.5 kV) to produce the primary ions. The sampling flow went into the instrument through a ~0.6-meter ¾ inch stainless steel tube. The sampling flow was 10 LPM and the sheath flow was set to 20 LPM. Data acquisitions for CI-APi-TOF was performed with a time resolution of 10 s. A calibration factor of $1.0 \times 10^{10}$ cm$^{-3}$ for SA was determined with sampling loss corrections before the campaign according to the method proposed by Kurten et al. (2012).

Dimethylamine (DMA) measurements were performed using a Vocus CI-ToF (time-of-flight) mass spectrometer (hereafter Vocus, Aerodyne Research Inc. & Tofwerk AG) using $H_3O^+$ as a reagent ion. The Vocus has been described in detail in Krechmer et al. (2018) and the study by Wang et al. (2020) utilized Vocus for DMA observations. In this study, the Focusing Ion-Molecular Reactor (FIMR) of Vocus operated at a pressure of 2.0 mbar and a temperature of 100 ℃ with the radio frequency amplitude of 350 V and frequency of $1.4 \times 10^6$ Hz. Data acquisition was performed with a time resolution of 10 s in the mass range 0 –1000 amu.

### 2.2.2 Particle size distribution measurements
**Particle Size Magnifier**

The Airmodus A11 nano-CNC-system (nano-Condensation Nucleus Counter), colloquially known as the Particle Size Magnifier (PSM) is a two-step condensation particle counter (CPC) capable of measuring particle size distributions of sub-3nm particles (Vanhanen et al., 2011). The system consists of two parts, in which the PSM (Airmodus A10) acts as a preconditioner where particles are grown first before being funneled to the CPC (Airmodus A20) for further growth and optical detection. In the PSM the sample flow is turbulently mixed with a heated flow saturated with diethylene glycol (DEG) in the mixing section and the DEG then condenses on the particles in the growth tube. By scanning the flow rate through the DEG saturator, the smallest activated particle size is altered which can be converted into a sub-3nm particle size distribution. Further particle growth is achieved by butanol in the CPC such that the particles reach optically detectable sizes.

The PSM was calibrated according to the standard operation procedure for PSM (Lehtipalo et al., 2022) using a known aerosol population from a glowing tungsten wire generator (Kangasluoma et al., 2015; Peineke et al., 2006). The detection efficiency for different particle sizes was determined by comparing the concentration of size selected particles to a reference instrument, in this case a Faraday cup electrometer.

The system was set up with an Airmodus Nanoparticle Diluter (AND) inlet (Lampimäki et al., 2023) for sample dilution and automatic background measurement to make sure that the CPC stays within a single counting range during the campaign. The inlet was set up at around 2 meters above the ground

and the background was measured roughly every 8 hours and subtracted from the signal during the
inversion process.

**HFDMPS and Hauke-type DMPS**

The high-flow differential mobility particle sizer (HFDMPS) system utilizes a half-mini differential
mobility analyzer (DMA, (Fernández De La Mora and Kozlowski, 2013; Cai et al., 2018)) to size-select
particles that are then grown and detected by an A11 nano Condensation Nucleus Counter system
(Airmodus Ltd., A11 nano-CNC) (Kangasluoma et al., 2018). The HFDMPS significantly improves
sub-10 nm particle measurements compared to a typical differential mobility particle sizer (DMPS)
system, allowing us to better characterize the sub-10 nm particle size distribution when combined with
the PSM measurements. The DMA was size-calibrated with electro sprayed positively charged
monomer ions of tetraheptylammonium bromide (THA+) (Ude and De La Mora, 2005).
The HFDMPS inlet was set up at a height of 1 m and used a 50 cm long 10 mm outer diameter tube
with a core sampling system to minimize losses (Kangasluoma et al., 2016; Fu et al., 2019). A home-
built Soft X-Ray ionization source (similar to the TSI Inc. Model 3087) was used to charge particles.
The HFDMPS measured the particle size-distribution from 2–15 nm for both polarities at 15 predefined
size-steps within 10 minutes.
Sampling from the same inlet and using the same charging device, a conventional DMPS system
equipped with a Hauke-type DMA (aerosol flow 1 LPM, sheath flow 5 LPM) and a TSI Inc. CPC
(Model 3772) was measuring the particle size-distribution from 10–800 nm at 16 predefined size-steps
within 10 minutes. In addition, a DMPS measuring from 15–800 nm was available in another
measurement container at the same field site. The total particle number concentrations obtained from
integrating the particle size-distribution measured by the DMPS was compared with a reference CPC
(TSI Inc. Model 3025A) operated at the same site during the first weeks of the campaign. It revealed
on average a factor of 2 lower concentrations measured by the Hauke-type DMPS which was confirmed
to be rather size-independent by a comparison of the measured size-distributions and their overlap with
the HF-DMPS system and was thus subsequently corrected for.

**2.2.3 Co-located measurements**

Additional co-located measurements of auxiliary data from CNR-ISAC network (www.isac.cnr.it/en)
and from the routine monitoring program of the Regional environmental protection agency of Emilia
Romagna (ARPAE, https://www.arpae.it/it) were used in this study. An online High-Resolution Time-
of-Flight Aerosol Mass Spectrometer (HR-ToF-AMS, Aerodyne Research) and a Multi Angle
Absorption Photometer (MAAP, Thermo Scientific) were operated on the same site for the
measurement of non-refractory species and black carbon (BC), respectively. Trace gases were also
measured with 1 minute time resolution: $O_3$ (Thermo Scientific, model TEI-49i), $NO_x$ (Teledyne-API,
model 200A), $NH_3$ (Teledyne-API, model 201E), and $SO_2$ (Thermo Scientific, Model 43i Trace Level-
Enhanced). Moreover, meteorological parameters (e.g., RH, temperature, wind direction and wind
speed) were measured by a meteorology station (VAISALA Ltd, model wxt536).

**2.3 Data processing**

**2.3.1 New particle formation classification**

We classified each day according to whether a growing mode appeared in the particle size distribution
or not. This classification was done separately for both the HFDMPS and the PSM size distributions. A
growing mode was defined as a new particle mode that appeared in the particle size distribution and
continued to grow to larger sizes for at least two hours. If there was a growing mode visible in both the
PSM and HFDMPS size distributions, the day was defined as "NPF with growth". If there was no
growth or the growth was unclear in the HFDMPS size distribution but there was a growing mode in
the PSM size distribution, then the day was classified as "NPF with no growth". If there was no growing
mode in either size distribution, then the day was marked as "no NPF events". The definition is similar
to Dada et al. (2018) who used naturally charged ions to separate between NPF days with clustering

only and clustering plus visible growth. If there was a growing or an undefined new mode visible in the combined size distribution but there was no clustering detected by the PSM, this day was marked as "unclear". Days that lacked data from one of the instruments were marked as "no data".

**2.3.2 Condensation sink, nucleation and growth rate calculations**

The condensation sink and coagulation sink were calculated according to Dal Maso et al. (2005) from the Hauke-type DMPS size distribution without any correction of aerosol hygroscopic behavior. Growth rates were calculated using the maximum concentration method, in which we fit a Gaussian distribution to the particle concentration evolution at a fixed size to determine the time of maximum concentration for a given size channel in the HFDMPS.

The growth rates were calculated by first determining the time to reach 50% of the maximum concentration and then the average growth rate is derived as the slope of the linear fit between the time and diameter:

$$GR = \frac{\Delta d_p}{\Delta t} \approx \frac{d_{p,f} - d_{p,i}}{t_f - t_i}, \qquad (1)$$

where $d_{p,f}$ is the diameter at the end time $t_f$ and $d_{p,i}$ is the diameter at the start time $t_i$.

From these, the growth rate was calculated as the slope of a linear least squares fit to the time-points of maximum concentration and their corresponding particle diameters. The formation rates were calculated for several sizes by using the balance equation in Kulmala et al. (2012) using the combined DMPS size-distributions ($J_2$, $J_3$, $J_6$) and the PSM plus combined DMPS size-distribution ($J_{1.7}$). Formation rates were then calculated by rearranging the equation describing the time evolution of the particle size distribution. Formation rate for a given diameter $d_{p1}$ is calculated as

$$J_{dp1} = \frac{dN_{dp1-dp2}}{dt} + CoagS_{dp1} \cdot N_{dp1-dp2} + \frac{GR}{\Delta d_p} N_{dp1-dp2}, \quad (2)$$

**2.3.3 Mass spectrometer data analysis**

The APi-TOF and CI-APi-TOF data were analyzed using the Tofware package (v.3.1.0, Tofwerk, Switzerland, and Aerodyne, USA) in the Igor Pro software (v.7.08, WaveMetrics, USA). The mass accuracy is within 10 ppm (APi-TOF) and 5 ppm (CI-APi-TOF), and the mass resolutions were ~4500 (APi-TOF) and ~5000 (CI-APi-TOF) for ions >200 Th. The raw signals were firstly normalized by the primary ions ($NO_3^-$, monomer, dimer and trimer) and then multiplied by the calibration factor of SA. Detailed information on the mass spectrometer data analysis methods can be found in previous studies (Cai et al., 2022; Cai et al., 2023a; Zha et al., 2018; Zha et al., 2023a; Fan et al., 2021; Zha et al., 2023b).

**2.3.4 Kinetic model Simulations**

In order to evaluate the contribution of SA-amine clustering to cluster formation in the Po Valley, we applied a kinetic model to simulate SA dimer concentrations. We simulated the cluster concentrations and particle formation rates under different amine levels based on the model. The simulation was performed with a temperature of 283 K, atmospheric pressure of $1.01 \times 10^5$ Pa, and the condensation sink (CS) of 0.01 s$^{-1}$ based on our measurement during the sampling period. In the model, the formation rate of SA tetramer was regarded as the simulated particle formation rate. The standard molar Gibbs free energy of formation and the corresponding evaporation of SA-amine clusters was based on quantum chemistry with corrections from the experimental data. The detailed settings of the kinetic model can be found in Cai et al. (2021).

**3. Results and discussions**

**3.1 NPF event frequency in Po Valley**

During the measurement period, frequent NPF events occurred in Po Valley (Fig. 2, Fig. S1). On 27% of the days, we observed new particle formation with growth at the site, while on 39% of the days we

observed new particle formation without growth (Fig. S1). In total we observed new sub-3 nm clusters forming on 66 % of the days. Even though we applied the similar definition of NPF events as previous study, we can only compare our NPF events with growth type with the reported NPF event frequency due to the lack of capacity to measure the sub-3nm particles in previous literature. Our results were similar to those by Hamed et al. (2007) who observed NPF events on 36 % of the time in March and April of 2002 at the same site. Manninen et al. (2010) observed NPF events during more than half of all days from March to Oct in 2008 and Kontkanen et al. (2016) observed NPF during 89 % of the days in July at the same site, which is higher than our observations. Hamed et al. (2007) also observed that NPF with growth events on 60% of the days during summer, which suggests that summertime NPF frequency at SPC is typically higher than our observation in springtime 2022. This difference in the observed NPF frequency was likely due to the different season with favorable conditions for NPF such as potential lower CS (due to less stagnant meteorological conditions) and higher basic and organic molecule concentrations in summer. In addition, the abundant solar radiation and low aerosol water content (limiting surface area and heterogenous reactions (Du et al., 2022)), likely create favorable conditions for NPF to occur.

The median average particle formation rates at 1.7 nm, 3 nm and 7 nm for all sampling days with NPF with growth events were 87 $cm^{-3}$ $s^{-1}$ (32 – 133 $cm^{-3}$ $s^{-1}$), 3.2 $cm^{-3}$ $s^{-1}$ (1.4 $cm^{-3}$ $s^{-1}$ – 7.0 $cm^{-3}$ $s^{-1}$) and 1.4 $cm^{-3}$ $s^{-1}$ (0.3 $cm^{-3}$ $s^{-1}$ – 3.0 $cm^{-3}$ $s^{-1}$), respectively. The formation rate at 1.7 nm during NPF with growth days (NPF with growth, 87 $cm^{-3}$ $s^{-1}$) is similar to that observed previously at the same site by Kontkanen et al. (2016) in summer. The high formation rate, which is comparable with heavily polluted urban environments such as Beijing and Shanghai, China (59 $cm^{-3}$ $s^{-1}$ – 225 $cm^{-3}$ $s^{-1}$ (Deng et al., 2020; Yao et al., 2018)), will be further discussed in section 3.4. The average formation rate ($J_{1.7}$) on NPF days without growth (24 $cm^{-3}$ $s^{-1}$) is much lower. During the noontime, the formation rate of particles for NPF events with no growth was less than half of $J_{1.7}$ for NPF with growth (Fig. S2). It suggests that for particles to grow in a polluted environment such as the Po Valley, there needs to be abundant clustering to overcome losses to the existing condensation sink so that at least some of the particles survive to grow into larger sizes.

SA has long been known as a primary gaseous precursor for NPF in continental environments, owing to its extremely low volatility (Kirkby et al., 2011; Kulmala et al., 2013). During our sampling period, we observed high SA concentration in the Po Valley, in accordance with the frequent NPF events. The daily average SA concentration measured between 10:00 – 14:00 LT was $4.6\times10^6$ $cm^{-3}$, which increased to $8.5\times10^6$ $cm^{-3}$ during NPF events with growth, aligning with previous findings from the same site ($1.6\times10^7$ $cm^{-3}$ during NPF in summer of 2009, (Paasonen et al., 2010)). Over the entire sampling period (10:00 – 14:00 LT), SA showed a moderately correlation with the calculated $J_{1.7}$ ($r$ = 0.49, Spearman correlation coefficient, for the logarithmic values), but its relationship varied among different days. This suggests that in addition to SA, other components, such as basic molecules, may also contribute to driving NPF events and subsequent growth in the Po Valley.

## 3.2 Nucleation mechanism

To investigate the NPF mechanism in the Po Valley, in this study we firstly compared the simultaneously measured $J_{1.7}$ and SA with recent Cosmics Leaving Outdoor Droplets (CLOUD) chamber experiments that simulated NPF under polluted boundary layer conditions with anthropogenic emissions (Xiao et al., 2021). In those experiments, amines, ammonia, as well as aromatics were added to reflect a heavily anthropogenic emission-influenced environment. Certain basic molecules, including amines (e.g., dimethylamine (DMA)) and ammonia ($NH_3$) have been shown to substantially enhance nucleation and reduce evaporation by stabilizing atmospheric SA in chamber studies (Almeida et al., 2013). Besides, OOMs can also contribute to NPF and subsequent particle growth, even without the inclusion of SA (Kirkby et al., 2016; Xiao et al., 2021). As shown in Fig. 3a, most of the measurements were above the SA-$NH_3$ system at 278K from the CLOUD chamber, suggesting the SA-$NH_3$ mechanism itself cannot solely explain the measured $J_{1.7}$ and that other species are most likely participating to NPF

in the Po Valley. For instance, amines, such as DMA or TMA, with higher basicity may contribute to NPF, consistent with not negligible concentrations of amines in previous studies in the aerosol at SPC (Paglione et al., 2014; Decesari et al., 2014). For the whole sampling period, the median SA and $J_{1.7}$ values in Po Valley follows the SA-DMA-NH$_3$ (4 ppt DMA and 1ppb NH$_3$) and SA-DMA-NH$_3$-Org (adding additional oxidized aromatic organics (Xiao et al., 2021)) lines from the CLOUD chamber at 293K even though during most of the NPF days the average noontime temperature was around 285K (Fig. 3a).

The SA dimer measured by CI-APi-TOF is typically used as an indicator for the initial step for the cluster formation in NPF events (Yan et al., 2021). According to a previous study (Yan et al., 2021), the source and sink terms of the SA dimer can be determined by calculating the formation rate from SA monomer collisions and the loss rate from the SA dimer through coagulation onto pre-existing particles (Fig. 2b). In general, the correlation coefficient between SA dimer and its source to sink term ratios ($r$ = 0.80, Spearman correlation coefficient) indicated that similar to Chinese urban areas, SA dimer was in a pseudo steady-state between the formation of SA monomer collision and the loss onto CS by coagulation.

To further assess the influence of DMA, one of the most common and efficient base molecules for NPF in urban environments (Yao et al., 2018), we compared the measured SA dimer concentrations with the simulated ones under different DMA levels (from 0.1 ppt to reaching kinetic limit) by the kinetic model (Fig. 3b). From our cluster kinetics simulations, during the peak hours of NPF, DMA concentrations are expected to be in the range of 0.1 ppt to 5 ppt, which is lower than the need for reaching the kinetic limit (Figs. 3b and S3). It implies that other factors, for example, the abundant ambient NH$_3$ concentrations (~10 ppb) or trimethylamine (TMA) during our study period may also participate in cluster formation. It is consistent with the Vocus measurement, which suggests the ambient DMA signals were close to the background levels (Fig. S4). The reason for not reaching SA-DMA limit during the campaign could be 1) the relatively lower DMA emissions (such as vehicle flows) than Chinese megacities (Ge et al., 2011; Zhu et al., 2022), and 2) the quickly scavenge caused by photolysis and nighttime high RH (85%) (Leng et al., 2015; Yao et al.,2016). Therefore, both of the abundant ambient NH$_3$ concentrations (~10 ppb) and amines likely participated in cluster formation during our study period.

Median particle growth rates (GR) during NPF events for 1.5 – 3 nm, 3 – 7 nm, 7 – 15 nm were 1.3 (1.0– 2.4) nm h$^{-1}$, 4.6 (2.9 – 5.8) nm h$^{-1}$, and 5.1 (3.8 – 8.8) nm h$^{-1}$, respectively. The values in brackets represent the 25$^{th}$ and the 75$^{th}$ percentile of data (Fig. 3c). Growth rates increase with particle diameters, a phenomenon observed in other campaigns around the world as well (Kontkanen et al., 2017, Kulmala et al., 2013)), typically indicative of an increasing organic vapors contribution with size (e.g., Stolzenburg et al. (2018)). The growth rates observed here were similar to those observed by Kontkanen et al. (2016) at SPC in summer (7.2 nm h$^{-1}$ for 7 – 20 nm) and our 1.5 – 3 nm growth rate matches well with Manninen et al. (2010) (1.5 nm h$^{-1}$) during spring in the Po Valley. A comparison to predicted growth rates from sulfuric acid condensation without organics, which was calculated based on kinetic collisions of the measured SA concentrations and the effect of van-der-Waals forces on the collision frequency ((Stolzenburg et al., 2020), Fig. 3c), suggests that sulfuric acid condensation may be on average sufficient for the growth of the smallest clusters. It supports the argument that in the initial steps of NPF and growth in Po Valley sulfuric acid and its stabilizing molecules (likely the bases NH$_3$ and amines) were controlling particle formation. However, for particles to grow beyond 3 nm in size other vapors were needed, which was suggested by the significantly lower contribution of growth by SA (indicated by the green line) than the measured GR for 3 – 7 nm and 7 – 15 nm (Fig. 3c). Those vapors were likely a mixture of organics of anthropogenic and biogenic origin (with the latter emitted at higher rates during summer, which could cause the slightly higher values in Kontkanen et al. (2017)). We compared the GR during NPF with and without growth events using the method proposed in Kulmala et al. (2022) where the signal was averaged for all classified non-event days and then an

appearance time fit was performed for each size channel independently, revealing also a growth pattern.
We found no significant difference for the GR in 7 – 15 nm size range (GR=5.1 nm h$^{-1}$ in NPF with
growth days and average GR=6.1 nm h$^{-1}$ in NPF without growth days). Considering the similar CS and
GR levels for NPF with and without growth days, the higher formation rates at 1.7 nm (87 cm$^{-3}$ s$^{-1}$) may
be a more important factor to surpass the CS. In stable meteorological conditions, a higher formation
rate may significantly elevate the possibility of newly formed particles overcome the CS and continuous
grow to larger sizes.

**3.3 Ion and neutral clusters and further particle growth**

During the campaign, we observed and identified different types of ion clusters with cluster ion
measurements using the APi-TOF, including SA-NH$_3$, SA-Amine, SA-NH$_3$-Amine, SA-NH$_3$-Org
during NPF. In Fig. 4a, we presented the mass defect plot of the naturally charged ion clusters on April
20$^{th}$, when strong NPF events were observed ($J_{1.7}$: 83 cm$^{-3}$ s$^{-1}$). The presence of these clusters was
usually in conjunction with SA tetramers (SA$_4$), pentamers (SA$_5$), and hexamers (SA$_6$), which
potentially contribute to the NPF events. In Api-TOF measurement, the absence of basic species in the
smallest sulfuric acid clusters is likely attributed to the loss of base molecules within the mass
spectrometer (Cai et al., 2022b; Zha et al., 2023; Alfaouri et al., 2022).
Among all SA-base (SA-B) clusters, the most abundant SA-NH$_3$ clusters were from SA$_4$-B to SA$_6$-B
(Fig. 4a), even though they are reported to be more easily evaporated than DMA clusters due to
collision-induced dissociation (Passananti et al., 2019). Pure SA-Amine clusters were only found in the
SA$_4$-B clusters with different types of amines, including methylamine (C$_1$-amine), DMA (C$_2$-amine),
trimethylamine (C$_3$-amine), and butylamine (C$_4$-amine). The detection of other SA-B than SA-DMA
clusters indicates that other candidate bases could also play a crucial role in the complex atmosphere
for nucleation. For example, a recent study conducted in Beijing highlights the importance of TMA,
which can enhance nucleation rate from SA-DMA system by 50% – 100% (Cai et al., 2023b). In the Po
Valley, the signal intensity of SA$_4$-NH$_3$ was significantly higher than that of the pure SA$_4$-amine clusters
(~2 times) even though amines (e.g., DMA) were proven to be more efficient (~3 orders of magnitude)
than NH$_3$ in clustering (Almeida et al., 2013). SA-NH$_3$-Amine clusters could be found along with SA-
NH$_3$ clusters in SA$_5$-B and SA$_6$-B. Similar patterns of the high fractions of SA−NH$_3$ and SA-NH$_3$-
Amine clusters were also reported in the CLOUD chamber studies under relatively low DMA and high
NH$_3$ conditions (Schobesberger et al., 2013). Therefore, it can be concluded that a large amount of NH$_3$
also participates in NPF in the Po Valley region. Meanwhile, with a much lower amount, amines may
also play a crucial role in the formation of small clusters (SA-B) due to their high stabilization
efficiencies.
Moreover, some SA-NH$_3$-Org and I-containing ion clusters were also observed on NPF days, but to a
much lower extent than clusters involving NH$_3$ or DMA. It has been shown in previous CLOUD
chamber studies that the oxidation products of anthropogenic volatile organic compounds (AVOCs, e.g.,
naphthalene, trimethylbenzene and toluene) can largely promote the formation rate of particles (Xiao
et al., 2021). The I-containing ions (mainly IO$_3^-$) likely originated from the Adriatic Sea during the
daytime, which was indicated by the easterly wind. Since no large iodine clusters were identified in the
APi-TOF (e.g., $(HIO_3)_{0-1}(I_2O_5)_n \cdot IO_3^-$, (He et al., 2021)), iodine-induced new particle formation in the
Po Valley may not be as important as the pristine marine environment (Sipila et al., 2016). During NPF
without growth days, the formation mechanism was similar to the NPF days regarding the ion cluster
measurement (Fig. S5).
The SA monomer in the Po Valley can be observed during the peak hours (10:00 − 14:00 LT) in both
NPF and non-NPF days, but much lower SA dimer or trimers were found in the non-NPF days (Figs.
4b, and S6). In the nighttime, the SA concentrations were close to zero due to the scavenging of SO$_2$
and SA by hydrated aerosol and hygroscopic growth of particles, as indicated by the high RH (Fig. 1).
During our sampling period, large amounts of organics were identified by the CI-APi-TOF. They were

typically smaller than 400 Th with carbon numbers < 8 and oxygen numbers < 6 (Fig. S7). Due to the relatively high $NO_x$ levels (13 ppb) that can terminate the dimerization reactions (Yan et al., 2020), no OOM dimers were found, which is different from clean and biogenically dominated environments such as Hyytiälä (Lehtipalo et al., 2018). The compositions of OOMs were similar between NPF and non-NPF days but with different abundance. Extremely high abundances of nitrophenols and their homologous compounds were found on non-NPF days (~8 times higher than on NPF days), likely caused by both of the enhanced primary (e.g., biomass burning (Mohr et al., 2013) and pesticide usage (Harrison et al., 2005)) and secondary (e.g., photochemical and/or aqueous-phase secondary formation) sources (Zheng et al., 2021; Gilardoni et al., 2016). $C_{2-4}H_{4,5}N_{0,1}O_{3,4}$ compounds were found to be 50% higher (Fig. S7) on non-NPF days due to the higher RH and the enhanced heterogeneous reactions that form smaller organics such as carboxylic acids. Previous studies also reported aqueous-phase organic aerosol processing at high RH (Gilardoni et al., 2016) and high concentrations of carboxylic acids such as formic, oxalic, and malonic acids in the springtime in the Po Valley (Saarikoski et al., 2012). In general, the fraction of the abundance of nitrogen-containing OOMs (CHON) of total identified OOMs were 60% − 70%, which is close to the levels reported in polluted cities such as Nanjing (Nie et al., 2022) and Beijing (Guo et al., 2022). A slightly higher fraction of CHON compounds (73 %) was found during non-NPF days than NPF days (67 %), consistent with higher NOx and fine particulate matter levels (Fig. S8). It is likely associated with the stagnant meteorological conditions and accumulation of pollutants during the non-NPF days. However, the overall high amounts of CHON compounds and the lack of organic dimers make it unlikely that OOMs drive the NPF process (both clustering and initial growth, see e.g., Simon et al. (2020)). Their similar abundance on non-NPF and NPF days was also in line with the similar estimated GR for both types of days.

Throughout the entire sampling period, relatively high concentrations of fine particulate matters ($PM_{2.5}$) were measured, with a daily average of 17 µg m$^{-3}$ and a maximum value of 43 µg m$^{-3}$. Correspondingly, the hourly CS levels, which quantify the ability of pre-existing particles to scavenge gaseous precursors, ranged from $<1 \times 10^{-4}$ s$^{-1}$ to $3 \times 10^{-2}$ s$^{-1}$ with an average value of $5.4 \times 10^{-3}$ s$^{-1}$. Previous studies in polluted areas, such as Chinese megacities, have shown that NPF events are closely linked to CS levels (Cai et al., 2017). NPF probability was reported to decreased to 50% when CS was around $1 \times 10^{-2}$ s$^{-1}$ and completely shut off with CS of $6 \times 10^{-2}$ s$^{-1}$ (Du et al., 2022). However, in the Po Valley, we observed no strong influence of CS on NPF events, with only a slightly difference in CS during the noontime of non-NPF days (median: $9.4 \times 10^{-3}$ s$^{-1}$) than NPF days (median: $8.6 \times 10^{-3}$ s$^{-1}$).

**3.4 Comparison between Po Valley and other environments**

Even though the measured $J_{1.7}$ in Po Valley was at the same level of the values found in Chinese polluted megacities, it was much higher than in clean environments, such as the boreal forest of Hyytiälä in Finland, mountain sites Jungfraujoch in Switzerland, and Chacaltaya in Bolivia (1.5 cm$^{-3}$ s$^{-1}$ – 2.0 cm$^{-3}$ s$^{-1}$, Fig. 5a). The average SA concentrations ($4.6 \times 10^6$ cm$^{-3}$, 10:00 – 14:00 LT) were comparable to the levels observed in polluted megacities in China (ranging from $3.9 \times 10^6$ cm$^{-3}$ to $7.4 \times 10^6$ cm$^{-3}$, Fig. 5c), but significantly higher than those in remote areas like Hyytiälä ($9 \times 10^5$ cm$^{-3}$) and the Jungfraujoch ($5 \times 10^5$ cm$^{-3}$). SA concentrations during NPF days ($8.6 \times 10^6$ cm$^{-3}$) in the Po Valley were twice as high as those on non-NPF days ($4 \times 10^6$ cm$^{-3}$). This difference may be linked to the significant variations (t-test, p<0.05) of $SO_2$ concentrations between NPF days (0.38 ppb) and non-NPF days (0.20 ppb). This contrasts with findings in Beijing, where similar or even higher levels of SA and $SO_2$ were observed during non-NPF days compared to NPF event days (Yan et al., 2021). The variations in $SO_2$ and SA concentrations in the Po Valley could possibly be attributed to differences of air masses, as indicated by higher RH on non-NPF days (53%) than on NPF days (38%) but similar temperature (NPF days: 288 K, non-NPF days: 287 K). On higher RH days, photochemistry may be suppressed, potentially reducing the formation of sulfuric acid and low volatile condensable vapors.

The overall CS in spring (median: $8.9\times10^{-3}$ s$^{-1}$) in the Po Valley was lower than that in other polluted cities ($1.5\times10^{-1}$ s$^{-1}$ – $2.0\times10^{-1}$ s$^{-1}$), but significantly higher than that in clean environments ($2.0\times10^{-4}$ s$^{-1}$ (Hyytiälä and Jungfraujoch) – $3.0\times10^{-3}$ s$^{-1}$ (Chacaltaya with the influence of volcanoes), Fig. 5e). Contrary to Beijing or Shanghai where CS levels and efficiencies are the dominant factors for the NPF process (Du et al., 2022), NPF events in Po Valley are not strongly dependent on the CS levels ($9.4\times10^{-3}$ s$^{-1}$ and $8.6\times10^{-3}$ s$^{-1}$ for non-NPF and NPF days, respectively), likely due to generally lower CS levels than the Asian megacities (Fig. S8). The strength of precursor sources and their accumulation in the Po-Valley region might thus be more important for NPF to occur than the overall pre-existing sink for those precursors.

The average PM$_1$ concentrations during the sampling period was around 8 µg m$^{-3}$, significantly lower than New Delhi (268 µg m$^{-3}$), Beijing (33 µg m$^{-3}$, (Li et al., 2019)) and Shanghai (30 µg m$^{-3}$, (Song et al., 2023), Fig. S9). The major chemical compositions in PM$_1$ in Po Valley were similar to those in Beijing and Shanghai, with organics, ammonium nitrate, and ammonium sulfate being the most abundant components. However, PM$_1$ compositions in New Delhi differed from Po Valley and megacities in China. In New Delhi, strong biomass burning emissions with a high abundance of primary organics (155 µg m$^{-3}$, 58%) suppressed NPF events during the daytime from January to February but led to nocturnal particle growth, which is not observed in other polluted areas (Mishra et al., 2023).

Even with similar levels of CS and total PM$_1$ concentrations (NPF: 6.3 µg m$^{-3}$ and non-NPF: 6.5 µg m$^{-3}$) observed during noontime in Po Valley, the concentration of NO$_3^-$ increased by 50% on non-NPF days compared to NPF days, higher than the increase of PM$_1$ (3.1%) as shown in Fig. S9. A lower CS efficiency on NPF days due to lower fraction of nitrate was reported to suppress the scavenge of NPF precursors in Beijing (Du et al., 2022), which may also have the similar influence in the Po Valley. The observed growth rate for 7 – 15 nm particles in the Po Valley was about 5.1 nm h$^{-1}$, comparable to other urban and remote sites (2.9 – 9.1 nm h$^{-1}$, Fig. 5f). The general similar growth rates among different types of environments were also reported in previous studies (Deng et al., 2020), which needs further investigation in future research.

For the basic gaseous precursors, the average concentration of NH$_3$ was ~10 ppb, which was in the same range as that found in the Chinese megacities (10 – 30 ppb) and much higher than that at remote sites (<0.1 ppb, Table S1). The high NH$_3$ can be attributed to agricultural activities such as fertilization, which were widely applied during springtime in the region. The strong interference of ammonia emitted from fertilization to NPF was also observed in Qvidja, an agricultural site in Southern Finland (Olin et al., 2022). During our sampling period, measured DMA were too close to the detection limit of the Vocus (Fig. S2), and lower than those observed in the Chinese megacities (10 – 40 ppt, Fig. 5d). In the spring season, DMA in the Po Valley cannot fully stabilize all atmospheric SA clusters and hence NPF is very sensitive to variations in the concentrations of the different stabilizers (NH$_3$, DMA, and as shown by our analysis likely only to a lower extent organics). This could explain the scattered correlation between the formation rate and SA concentrations on different days (Fig. 3).

Therefore, in the Po Valley region, the initial nucleation of frequent NPF is primarily attributed to high sulfuric acid concentrations and basic molecules, including ammonia and various amines. This mechanism is generally similar to what is observed in Chinese megacities. However, in the Po Valley region, DMA, a typical base in anthropogenic emission-influenced areas, is insufficient to stabilize the high levels of sulfuric acid, leading to the involvement of other basic molecules like additional other type of amines and ammonia, likely originating from fertilization in the area. This involvement of ammonia and other amines differs from Chinese megacities such as Shanghai, where high levels of DMA were observed (~40 ppt, (Yao et al., 2018; Yao et al., 2016)). As insufficient DMA is available to stabilize all clusters, we speculate that the clustering is therefore sensitive to the abundance of amines and the variations in DMA or other amine concentrations would result in different formation rates. In that sense, during our sampling period, NPF in Po Valley seems to be more sensitive to the strength of certain emission sources of amines compared to megacity environments, where the clustering is "saturated" with respect to DMA (i.e., proceeding at the maximum kinetically possible rate). The

abundant OOMs dominate the consecutive growth process, leading to a comparable GR to Chinese megacities such as Beijing and Shanghai. Due to the relatively lower CS than these megacities, the newly formed particles may however have a higher survival probability compared to the megacities and provide more long-term surviving particles in the Po Valley, indicating a decisive role of NPF for Po-Valley aerosol and $PM_{2.5}$ concentrations.

## 4. Conclusions

In this study, we conducted a continuous two-month measurement campaign in the Italian Po Valley during springtime, where frequent NPF events were observed on 66% of all days. Through direct ion cluster measurement, kinetic models, and the comparison with the CLOUD chamber experiment, we have determined that sulfuric acid-base nucleation is the dominant formation mechanism in the Po Valley region. Abundant sulfuric acid and basic molecules, including amines and ammonia derived from agriculture activities, provided ample precursors for NPF events. In contrast to megacity environments, CS showed no significant difference between NPF event and non-event days, indicating that in Po Valley it is more the abundance of precursors than the variations in the sink controlling the occurrence of NPF. Furthermore, we observed that apart from DMA, a typical basic precursor, $NH_3$ and other amines were also likely to be involved in NPF in the Po Valley. This was supported by the high abundance of SA-$NH_3$ and SA-amine-$NH_3$ clusters measured by the APi-TOF during NPF events. DMA, while more efficient than ammonia, was insufficient to stabilize all SA during our sampling period. This resulted in a more scattered correlation between sulfuric acid concentrations and measured formation rates compared to Chinese megacities. In that sense, we could show that the clustering during NPF is clearly distinct between polluted megacity environments and polluted semi-urbanized regions such as Po Valley. Similar to Beijing, we found that OOMs did not play a decisive role in the initial cluster formations, likely due to the absence of ultra-low volatility organics (typical OOM dimers) in the ions and neutral cluster measurements. However, low-volatility organics were abundant enough to induce fast growth processes above 3 nm. The comparable GR and formation rates, along with lower efficient CS compared to megacity environments, indicate a high survival probability for the newly formed particles. Therefore, NPF is likely to play an important role in the fine particle concentrations and pollution levels in the Po Valley region. Further reductions of key NPF species, including $SO_2$, amines and $NH_3$, can contribute to suppressing NPF event frequency and lowering particle numbers. This, in turn, would improve air quality in the Po Valley region.

**Data availability**

Data are available from the authors upon request.

**Competing interests**

At least one of the (co-)authors is a member of the editorial board of Atmospheric Chemistry and Physics

**Author contributions**

JC, DS, FB, and MK designed the research. JC, JS, YFG, SH, MP, AN, FM, SD, MR, NZ and CM collected the data at the SPC site. JC, JS, YG, ST, RY, DA, QZ, DS and FB interpreted the data. MP, WH, YL, GC, LQ, KL, YG, CW, WN, JK, CM, QZ, DS, FB helped to improve the manuscript. JC, JS, DS, and FB wrote the manuscript with contributions from all co-authors. All authors have given approval to the final version of this manuscript.

**Acknowledgements**

The work is supported by the Academy of Finland (Center of Excellence in Atmospheric Sciences, project no. 307331, PROFI3 funding no. 311932, and ACCC Flagship no. 337549), the European Research Council via ATM-GTP (no. 742206), Consolidator grant INTEGRATE (no. 865799) and

CHAPAs (no. 850614), the European Union's Horizon 2020 research and innovation programme (project FORCeS under grant agreement no. 821205, H2020-INFRAIA-2020-1 grant agreement no. 101008004, Marie Skłodowska–Curie grant agreement no. 895875 (NPF-PANDA), the Vienna Science and Technology Fund (WWTF) through project VRG22-003, Jenny and Antti Wihuri Foundation, and the Knut and Alice Wallenberg Foundation (WAF project CLOUDFORM, grant no. 2017.0165). The authors also would like to thank the effort from all the researchers in the SPC site. The authors would also like to thank Chenjuan Deng, Mao Xiao and Lubna Dada for providing the supporting data in Beijing and CLOUD chamber experiment.

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

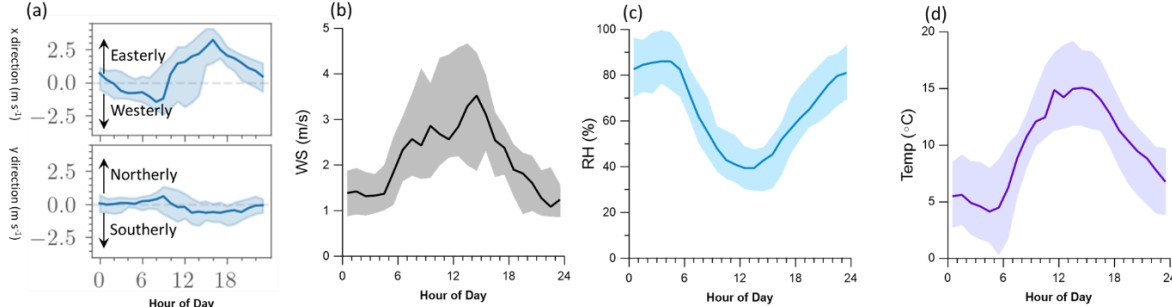


**Figure 1**. The diurnal variations of (a) average wind vectors, (b) wind speed, (c) relative humidity (RH), and (d)
temperature.


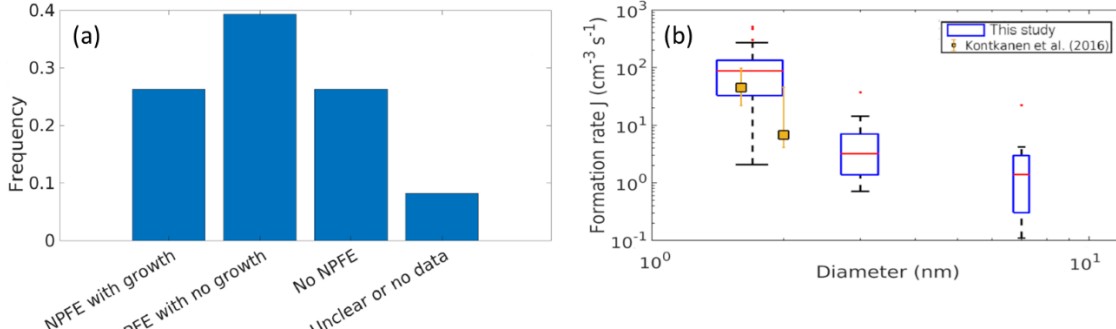


**Figure 2.** (a) The frequency of NPF events with and without growth, of days without NPF, and days with unclear
classification or no data during this study, (b) calculated formation rates at 1.7 nm, 3 nm and 7 nm from this
study and values reported by Kontkanen et al. 2016 (yellow squares). The red lines are the median values of the
maximum formation rates measured during an NPF event, the blue boxes show the values between 25th and 75th
percentiles and the black whiskers mark the 5[th] and 95[th] percentiles. Red dots are outliers, and the width of the
box is proportional to the square root of the number of the $J$ values.

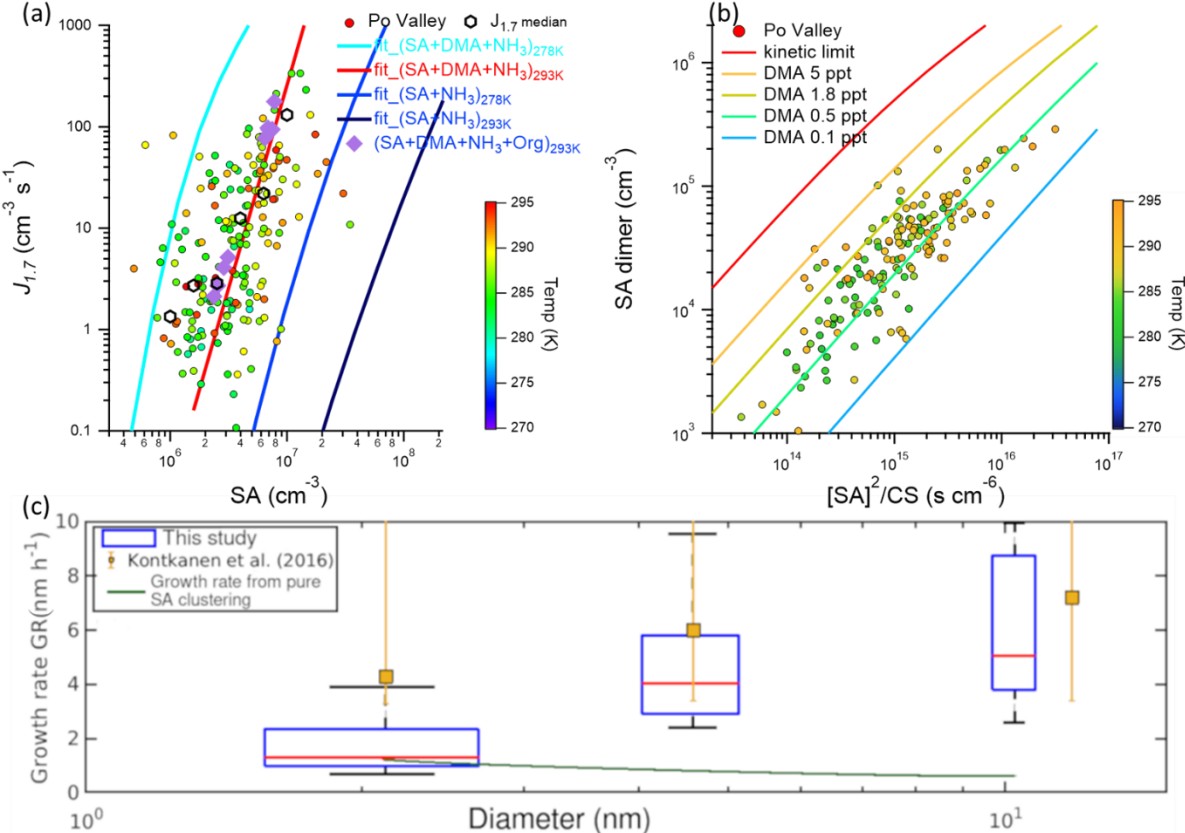


**Figure 3.** (a) The formation rate of 1.7 nm particles ($J_{1.7}$) versus SA concentrations in during springtime in the Po
Valley (shown as circles) and experimental results from CLOUD chamber experiments (shown as solid diamonds).
The solid lines are from fitted results of CLOUD chamber experiments and the black hexagon represented the
mean values under different SA levels, (b) the relationship between sulfuric acid dimer concentration (SA dimer),
the square of monomer concentrations ($SA)^2$, and the CS. The lines are from the kinetic model simulations under
different DMA levels and the dots are from the measurement. In (a) and (b), the results from the field
measurements are from the daytime (10:00 – 14:00 LT) and color-coded by the temperature at the site. The $J_{1.7}$
and corresponding SA concentrations of CLOUD chamber results are from previous literature (Xiao et al., 2021).
(c) Calculated growth rates for 1.5 – 3 nm, 3 – 7 nm, and 7 – 15 nm from this study and values reported by
Kontkanen et al. (2016, yellow squares). The red horizontal lines are the median values, the blue boxes show the
values between 25[th] and 75[th] percentiles and the black whiskers mark the 5[th] and 95[th] percentiles. The green solid
line represents predicted growth rates from pure sulfuric acid without organics condensation (Stolzenburg et al.,
2020). The width of the box is proportional to the square root of the number of the GR values.

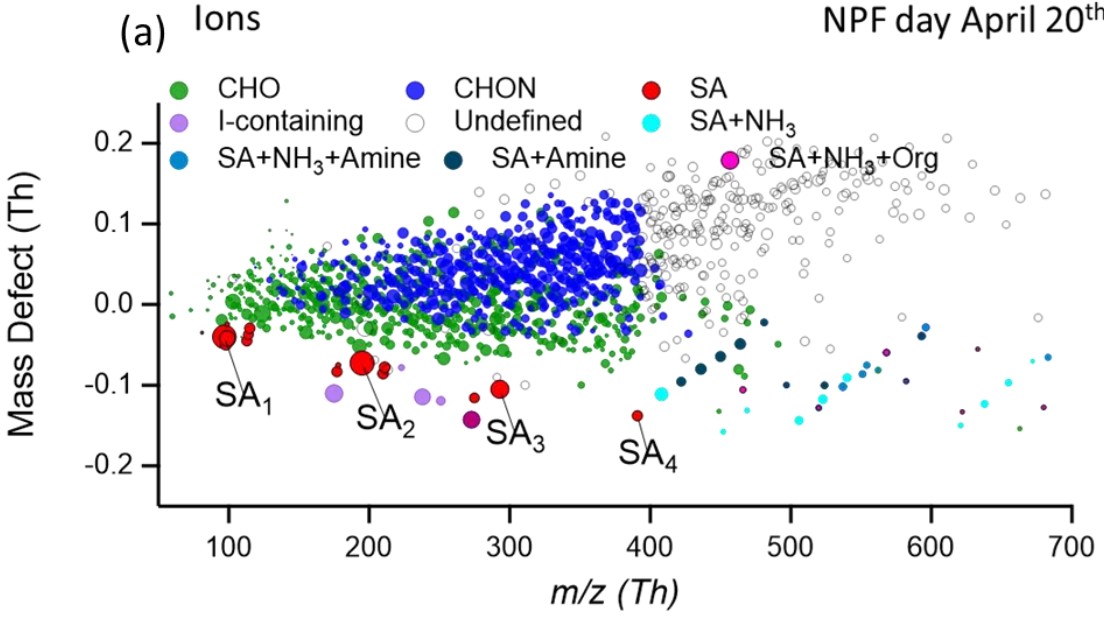

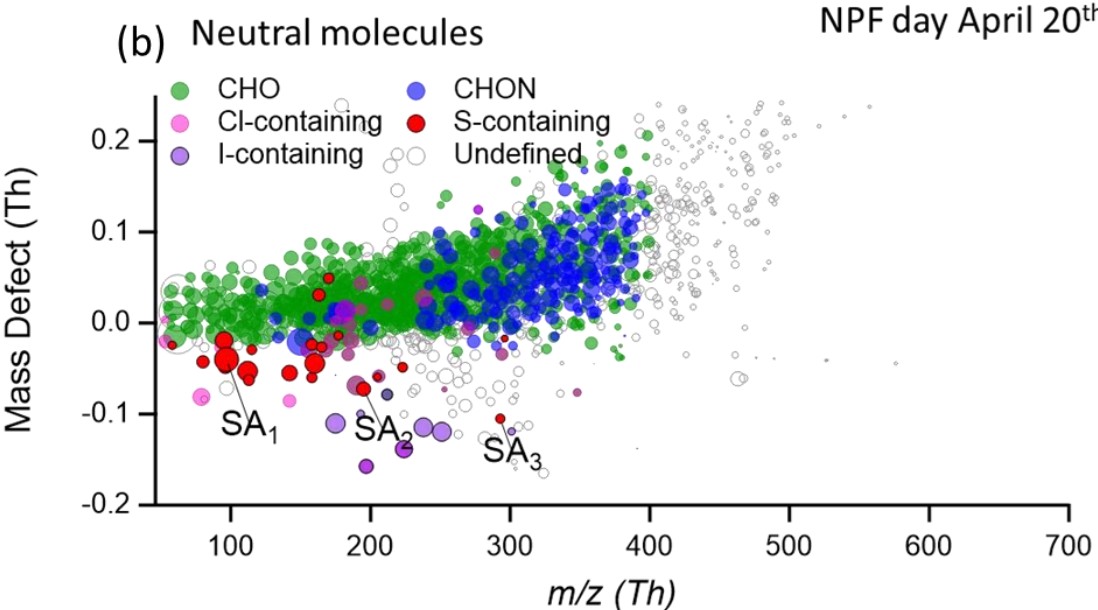

**Figure 4.** Mass defect plots, which represent the difference between compounds' exact mass and nominal mass, for (a) ion clusters and (b) neutral clusters during the NPF period (10:00 − 14:00 LT) of April 20. The size of the dots is proportional to the logarithm of the signal intensity of each cluster.

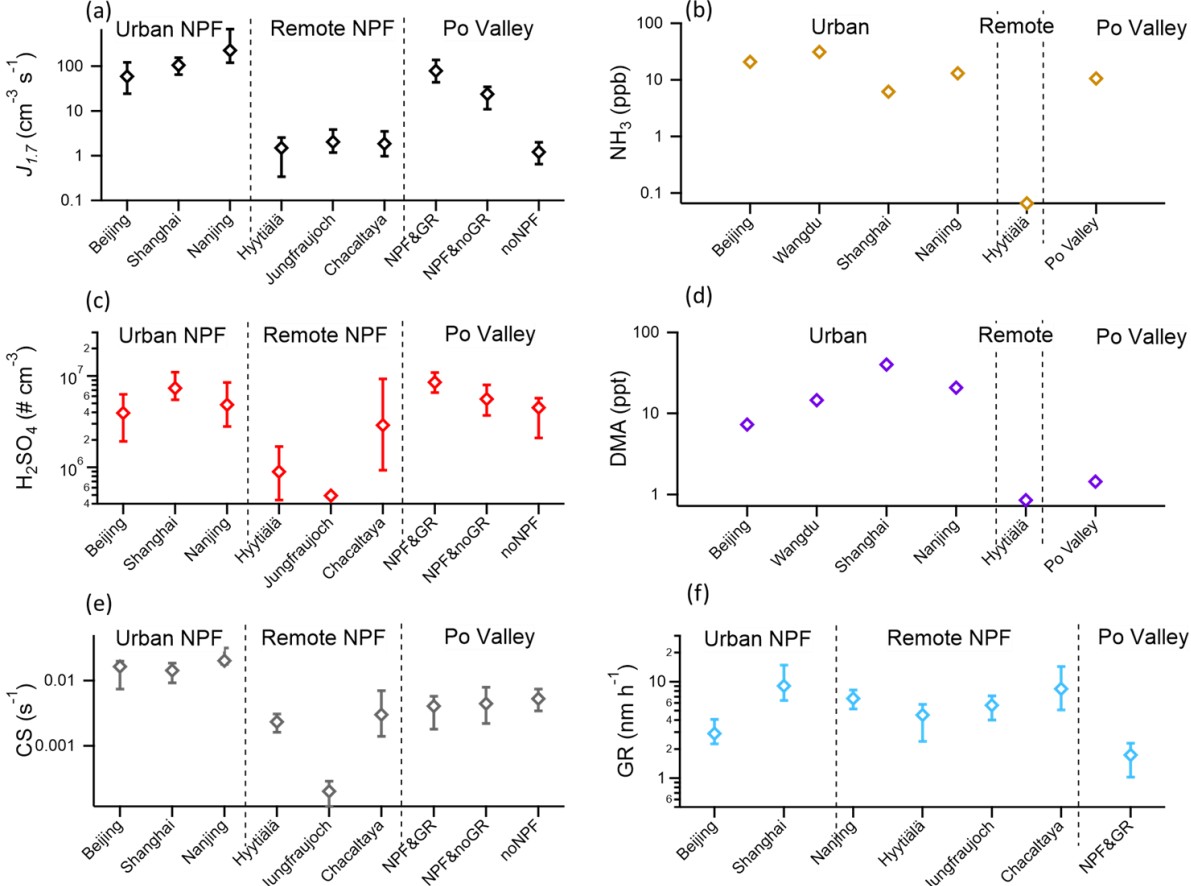

1076

**Figure 5.** Parameters and gaseous precursors of NPF in the Po Valley and other environments. (a) formation rate of sub-2 nm particles, (b) the atmospheric $NH_3$ concentrations, (c) SA concentrations, (d) DMA concentrations, (e) CS levels, and (f) growth rate in different environments. The diamond dots represent the median values, and the error bars represent the $25^{th}$ and $75^{th}$ percentiles. For the Po Valley data, the formation rates, growth rates, SA concentrations and CS data were selected for 10:00 – 14:00 LT. The formation rates, growth rates, SA concentrations and CS during NPF in Beijing, Shanghai, Hyytiälä, Jungfraujoch and Chacaltaya are from Deng et al. (2020). The GR calculation range varies for different sites. Beijing ($GR_{7-15}$, (Deng et al., 2020)), Shanghai ($GR_{7-25}$, (Yao et al., 2018)), Nanjing ($GR_{3-20}$, (Yu et al., 2016)), Hyytiälä ($GR_{3-20}$, (Vana et al., 2016)), Jungfraujoch ($GR_{7-20}$, (Boulon et al., 2010)), Chacaltaya ($GR_{7-20}$, (Rose et al., 2015)), and Po Valley ($GR_{7-15}$, this study) are used for comparison. The $NH_3$ and DMA concentrations are from literature, which is listed in the Table S1. Half of the limit of detection (LOD) of DMA concentrations in Hyytiälä was applied in panel d. DMA concentrations in Po Valley was not presented since it is not quantified in this study.