# Peer review of "polluted Po Valley, Italy"

_EGUsphere, 2023_

## Referee Comment (RC1)

Elucidating the mechanisms of atmospheric new particle formation in the highly polluted Po Valley, Italy. New particle formation (NPF) is a significant source of aerosol particles and cloud condensation nuclei in the troposphere, playing an essential role in air quality and climate. Frequent NPF events have been observed in heavily polluted urban environments, contributing to the aerosol number concentration by a significant amount. This topic is suitable for publication in this journal. However, the manuscript needs an improvement before publication. Therefore, I would like to recommend publication with 'major revision.'

1. Line 34: What exactly does organic refer to? The scope of organic is too large and needs to be clarified.

2. Line 30-41: More quantitative results need to be given. In lines 30-41, the quantitative results of this part are less, and the highlighted conclusion of this article is not given. Need to clarify the characteristics of NPF in the highly polluted environment? What are the mechanisms? For example, in lines 35-37, what was obtained using ion cluster measurements and kinetic model results? This statement is currently too general.

3. Introduction: This section needs to be reorganized. The summary of global NPF researches in the second paragraph is insufficient, especially the summary of NPF-related researches in polluted atmospheric environments. The importance and uniqueness of Po Valley is not adequately explained in the third paragraph. What makes Po Valley unique compared to other polluted areas? What are the differences, especially compared with relevant researches in China and India? In addition, what is special about the so-called highly polluted environment selected for this study? How does the highly polluted environment compare to studies in China and India? What are the existing NPF formation and growth mechanisms in polluted environments? How does this study differ from these previous studies?

4. In Section 2.3.2, Condensation sink, nucleation and growth rate calculations should give the calculation formula.

5. Sections 3.1 and 3.2 are merged.

6. Section 3.3: It is necessary to add some in-depth analysis of mechanisms. For example, what are the commonalities compared with research results in Shanghai, Beijing and New Delhi? What are the differences in the Po Valley's meteorological conditions, chemical compositions and aerosol background concentrations compared with these polluted areas? What impact do these differences have on the NPF incident? Is the growth mechanism of new particles in the NPF event in Po Valley similar to that in other polluted environments? What are the effects of being highly polluted?

7. The discussion in Section 3.4 is too simple and requires in-depth analysis to compare the similarities and differences in the generation and growth mechanisms under different atmospheric environments. In particular, the unique results of Po Valley need to be highlighted instead of simply comparing the differences in data values as currently done.

8. Line 268, "concentrations" should be "concentration." Pay attention to other similar singular and plural expressions in the manuscript.

9. Line 259, Please add citations for Beijing and Shanghai (59 cm-3 s-1 − 225 cm-3 s-1),

10. Line 344, The unit expressions in "GR=6.1 nm h-1" and "(1.0– 2.4) nm/h, 4.6 (2.9 – 5.8) nm/h, and 5.1 (3.8 – 8.8) nm/h" in line 323 should be consistent, which should also be noted in other parts of the article.

11. Conclusion. The authors should give unique conclusions on the formation and growth

mechanism of NPF in the highly polluted environment of Po Valley, especially compared with other polluted environments. Needs to focus on giving quantitative results.

---

## Author Comment (AC1)

We thank the reviewer for the helpful and constructive comments, which have led to significant improvements in the manuscript. We have carefully revised the text. Our point-by-point replies are given below (blue), following the referees' comments (black). Changes to the manuscript are marked with green and underlines. In our replies, the line numbers refer to the revised manuscript.

**Reviewer 4:**
New particle formation (NPF) can play an important role in urban smog formation, and thus has a considerable impact on air quality and public health. However, it is still unclear when and where sulfuric acid nucleation will cease to dominate urban particle formation as sulfur emissions continue to decrease. This study investigated the characteristics and mechanisms of frequently occurring NPF events in the polluted Po Valley region through a 2-month measurement using a suite of mass specs and particle sizers. The authors first calculated particle formation and growth rates of NPF events based on the measured particle number size distributions. In addition, the authors discussed the role of measured gas-phase sulfuric acid, highly oxygenated organics, and base compounds (ammonia and amines) in particle nucleation growth, through comparisons with results from the CLOUD chamber experiments. Finally, the author concluded that sulfuric acid and base compounds remain the primary driver of the frequent NPF, while oxygenated organics may play a more important in particle growth.

The research topic of this paper is novel and has important implications of interest to a broad range of atmospheric scientists, the measurement techniques are state-of-the-art, and the results are comprehensive. Overall, this is a relevant study that fits well within the scope of the ACP. However, the way the results are interpreted and discussed needs minor revision to improve the clarity for non-specialist readers. Here are my comments:

Response: We appreciate the reviewer's positive feedback on manuscript, and we carefully revised the manuscript accordingly to the reviewer's comments.

The key point: My understanding is that the lower levels of condensation sink and amine concentration are the two main differences in the Po Valley, compared to previous urban measurements, such as those taken in China. And interestingly, these two factors offset each other, resulting in somewhat comparable particle formation rates. This is actually a very important message that emphasizes the need to further reduce amine emissions as we reduce aerosol pollution in both Europe and China. These differences were discussed in the text, but were less clear in the abstract. I would suggest that the author modify the abstract and conclusions extensively to highlight the key point of this work.

Response: Thanks for the reviewer's suggestion for the highlights of this work. We have addressed the unique environment in Po Valley area than Chinese megacities, such as lower condensation sink and metrological conditions in the introduction section (please refer to the response to Reviewer3, Comment 3). The importance of reducing key species to suppress secondary particle formations is also added in the abstract and conclusion section as suggested by the reviewer.

Abstract

Line 31-42

"Our results demonstrate that in this area, frequent NPF events (66% of all days during the measurement campaign) are primarily driven by abundant sulfuric acid ($8.5\times10^6$ cm$^{-3}$) and basic molecules. In contrast, oxygenated organic molecules from the atmospheric oxidation of volatile organic compounds (VOCs) appear to have a minor role in the initial cluster formation but contribute significantly to the consecutive growth process. Regarding basic molecules, amines, typical species contributing to NPF in highly polluted areas such as Chinese megacities, are insufficient to stabilize all sulfuric acid clusters. Ion cluster measurements and kinetic models suggest that ammonia (10 ppb) must therefore also play a role in the nucleation process. Generally, the high formation rates of sub-2 nm particles (87 cm$^{-3}$ s$^{-1}$) and nucleation mode growth rates (5.1 nm h$^{-1}$) together with the relatively low condensational sink ($8.9\times10^{-3}$ s$^{-1}$) will result in a high survival probability of newly formed particles, making NPF crucial for the springtime aerosol number budget. Our results also indicate that reducing key pollutants such as SO$_2$,

amine and NH$_3$, could substantially help to decrease the particle number concentrations in the Po Valley region."

Conclusions

Line 521-544

"In this study, we conducted a continuous two-month measurement campaign in the Italian Po Valley during springtime, where frequent NPF events were observed on 66% of all days. Through direct ion cluster measurement, kinetic models, and the comparison with the CLOUD chamber experiment, we have determined that sulfuric acid-base nucleation is the dominant formation mechanism in the Po Valley region. Abundant sulfuric acid and basic molecules, including amines and ammonia derived from agriculture activities, provided ample precursors for NPF events. In contrast to megacity environments, CS showed no significant difference between NPF event and non-event days, indicating that in Po Valley it is more the abundance of precursors than the variations in the sink controlling the occurrence of NPF.

Furthermore, we observed that apart from DMA, a typical basic precursor, NH$_3$ and other amines were also likely to be involved in NPF in the Po Valley. This was supported by the high abundance of SA-NH$_3$ and SA-amine-NH$_3$ clusters measured by the APi-TOF during NPF events. DMA, while more efficient than ammonia, was insufficient to stabilize all SA during our sampling period. This resulted in a more scattered correlation between sulfuric acid concentrations and measured formation rates compared to Chinese megacities with higher DMA concentrations. In that sense, we could show that the clustering during NPF is clearly distinct between polluted megacity environments and polluted semi-urbanized regions such as Po Valley. Similar to Beijing, we found that OOMs did not play a decisive role in the initial growth processes, likely due to the absence of ultra-low volatility organics (typical OOM dimers) in the ions and neutral cluster measurements. However, low-volatility organics were abundant enough to induce fast growth processes above 3 nm. The comparable GR and formation rates, along with lower efficient CS compared to megacity environments, indicate a high survival probability for newly formed particles. Therefore, NPF is likely to play a significant role in the fine particle concentrations and pollution levels in the Po Valley region. Further reductions of key NPF species, including SO$_2$, amines and NH$_3$, can contribute to suppressing NPF event frequency and lowering particle numbers. This, in turn, would improve air quality in the Po Valley region."

Other amines: In Fig. 3(b), most of the data points fall between DMA concentrations of 0.5 -1.8 pptv, which is close to the measured DMA concentration in Fig. 5(d). Is there any other evidence to support the conclusion that DMA is not sufficient? Or are there other amines measured by the H3O+ CIMS? If there is no strong evidence, I would suggest that the authors tone down this assertion.

Response:

We draw the conclusion that DMA alone is not sufficient to stabilize all sulfuric acids based on the following observations:

1) In the ion cluster measurement by Api-TOF, if DMA is enough to stabilize all sulfuric acids (e.g., DMA=10 ppt), as indicated by the recent CLOUD chamber experiments, a significant fraction of SA should cluster with DMA given its stronger basic properties, and SA-NH$_3$ clusters can be neglected in the ion cluster measurement (Yin et al., 2021). However, our measurement does not align with this expectation (Fig. 4a).

2) As mentioned by the reviewer, in Fig 3b, SA-DMA cluster formation rates do not reach the kinetic limit during our measurement, indicating that gaseous DMA is insufficient to stabilize all SA during NPF events.

3) According to co-located Vocus-CIMS measurement, ambient DMA signals were close to background levels, despite the detection of a DMA peak in the mass spectra (Fig. S4).

We think other amines, such as TMA may also involve since the present of SA and other amine clusters in the Api-TOF measurement during NPF events, which were further discussed in the section 3.3.

Additionally, based on previous studies at the same site, other amines (e.g., methylamine and trimethylamine) were also observed in the particulate phase (Decesari et al., 2014; Paglione et al., 2014). Therefore, other gaseous amines were also likely involved in the NPF.

However, we agree in general with the reviewer that the discussions on the involvement of other amines and the lack of DMA should be tonged down due to the lack of quantitative gas-phase amine measurement in our campaign. In this study, our gas-phase amine measurement lacks quantitativeness due to the absence of a suitable calibration method and the Vocus-CIMS needs to be specially tuned for good performance with amines (Wang et al., 2020), which was not applied in our campaign. We, therefore, revised the manuscript as follows:

Line 340-346

"It implies that other factors, for example, the abundant ambient $NH_3$ concentrations (~10 ppb) during our study period may also participate in cluster formation. It is consistent with the Vocus measurement, which suggests the ambient DMA signals were close to the background levels (Fig. S4). The reason for not reaching SA-DMA limit during the campaign could be 1) the relatively lower DMA emissions (such as vehicle flows) than Chinese megacities (Ge et al., 2011; Zhu et al., 2022), and 2) the quickly scavenge caused by photolysis and nighttime high RH (85%) (Leng et al., 2015; Yao et al.,2016)."

Line 344: "…the higher formation rates at 1.7 nm (87 cm-3 s-1) may be the decisive factor to overcome the CS and determine if a growing mode can be observed leading to a classification of the day as an NPF with growth day." This is more speculation than fact – other factors such as meteorological conditions may also play an important role. I would suggest that the authors tone down this assertion.

Response:

We have revised the sentence as follows:

Line 371-375

"Considering the similar CS and GR levels for NPF with and without growth days, the higher formation rates at 1.7 nm (87 cm$^{-3}$ s$^{-1}$) may be a more important factor to surpass the CS. In stable meteorological conditions, a higher formation rate may significantly elevate the possibility of newly formed particles overcome the CS and continuous grow to larger sizes."

Line 352: "Acid-base clusters were not observed in monomer (SA1), dimer (SA2), or trimers (SA3), likely due to declustering effects in the APi-TOF instrument". This is not true – pure SA3 is energetically more stable than SA-base clusters.

Response:

The sentence has been revised as follows to make the statement clear:

Line 382-384

"In Api-TOF measurement, the absence of basic species in the smallest sulfuric acid clusters is likely attributed to the loss of base molecules within the mass spectrometer (Cai et al., 2022b; Zha et al., 2023; Alfaouri et al., 2022)."

Reference

Decesari, S., Allan, J., Plass-Duelmer, C., Williams, B. J., Paglione, M., Facchini, M. C., O'Dowd, C., Harrison, R. M., Gietl, J. K., Coe, H., Giulianelli, L., Gobbi, G. P., Lanconelli, C., Carbone, C., Worsnop,

D., Lambe, A. T., Ahern, A. T., Moretti, F., Tagliavini, E., Elste, T., Gilge, S., Zhang, Y., and Dall'Osto, M.: Measurements of the aerosol chemical composition and mixing state in the Po Valley using multiple spectroscopic techniques, Atmospheric Chemistry and Physics, 14, 12109-12132, 10.5194/acp-14-12109-2014, 2014.

Paglione, M., Saarikoski, S., Carbone, S., Hillamo, R., Facchini, M. C., Finessi, E., Giulianelli, L., Carbone, C., Fuzzi, S., Moretti, F., Tagliavini, E., Swietlicki, E., Eriksson Stenström, K., Prévôt, A. S. H., Massoli, P., Canaragatna, M., Worsnop, D., and Decesari, S.: Primary and secondary biomass burning aerosols determined by proton nuclear magnetic resonance (1H-NMR) spectroscopy during the 2008 EUCAARI campaign in the Po Valley (Italy), Atmospheric Chemistry and Physics, 14, 5089-5110, 10.5194/acp-14-5089-2014, 2014.

Wang, Y., Yang, G., Lu, Y., Liu, Y., Chen, J., and Wang, L.: Detection of gaseous dimethylamine using vocus proton-transfer-reaction time-of-flight mass spectrometry, Atmospheric Environment, 243, 10.1016/j.atmosenv.2020.117875, 2020.

---

## Author Comment (AC2)

We thank the reviewer for the helpful and constructive comments, which have led to significant improvements in the manuscript. We have carefully revised the text. Our point-by-point replies are given below (blue), following the referees' comments (black). Changes to the manuscript are marked with green and underlines. In our replies, the line numbers refer to the revised manuscript.

**Reviewer 3:**
Elucidating the mechanisms of atmospheric new particle formation in the highly polluted Po Valley, Italy. New particle formation (NPF) is a significant source of aerosol particles and cloud condensation nuclei in the troposphere, playing an essential role in air quality and climate. Frequent NPF events have been observed in heavily polluted urban environments, contributing to the aerosol number concentration by a significant amount. This topic is suitable for publication in this journal. However, the manuscript needs an improvement before publication. Therefore, I would like to recommend publication with 'major revision.'

Response: We appreciate the reviewer's positive feedback on manuscript and have revised the manuscript accordingly to the reviewer's comments.

1. Line 34: What exactly does organic refer to? The scope of organic is too large and needs to be clarified.

Response: now it has been changed to "oxygenated organic molecules from the atmospheric oxidation of volatile organic compounds (VOCs)" (Line 33-34) to clarify that oxygenated organic vapors were the main contributor to the growth of the newly formed particles.

2. Line 30-41: More quantitative results need to be given. In lines 30-41, the quantitative results of this part are less, and the highlighted conclusion of this article is not given. Need to clarify the characteristics of NPF in the highly polluted environment? What are the mechanisms? For example, in lines 35-37, what was obtained using ion cluster measurements and kinetic model results? This statement is currently too general.

Response:

To the best of our knowledge, our study, for the first time, elucidates that new particle formations in Po Valley is primarily caused by high atmospheric sulfuric acid and base, specifically amines and ammonia. Due to the limited concentration of dimethylamine (DMA), high concentrations of ammonia and other amines are likely to play important roles in the nucleation during the sampling period. This differs from the Chinese megacities such as Shanghai, where DMA emissions are abundant (Yao et al., 2016). Meanwhile, oxygenated organics molecules do not appear to play a crucial role in the nucleation process but significantly contribute to the growth process.

To better convey our main conclusion, we have incorporated the following changes in the revised manuscript as suggested:

Line 31-42

"Our results demonstrate that in this area, frequent NPF events (66% of all days during the measurement campaign) are primarily driven by abundant sulfuric acid ($8.5 \times 10^6$ $cm^{-3}$) and basic molecules. In contrast, oxygenated organic molecules from the atmospheric oxidation of volatile organic compounds (VOCs) appear to play a minor role in the initial cluster formation but contribute significantly to the consecutive growth process. Regarding alkaline molecules, amines, are insufficient to stabilize all sulfuric acid clusters in Po Valley. Ion cluster measurements and kinetic models suggest that ammonia (10 ppb) must therefore also play a role in the nucleation process. Generally, the high formation rates of sub-2 nm particles (87 $cm^{-3}$ $s^{-1}$) and nucleation mode growth rates (5.1 nm $h^{-1}$) together with the relatively low condensational sink ($8.9 \times 10^{-3}$ $s^{-1}$) will result in a high survival probability of newly formed particles, making NPF crucial for the springtime aerosol number budget. Our results also indicate that reducing key pollutants such as $SO_2$, amine and $NH_3$, could help to decrease the particle number concentrations substantially in the Po Valley region."

3. Introduction: This section needs to be reorganized. The summary of global NPF research in the second paragraph is insufficient, especially the summary of NPF-related researches in polluted atmospheric environments. The importance and uniqueness of Po Valley is not adequately explained in the third

paragraph. What makes Po Valley unique compared to other polluted areas? What are the differences, especially compared with relevant researches in China and India? In addition, what is special about the so-called highly polluted environment selected for this study? How does the highly polluted environment compare to studies in China and India? What are the existing NPF formation and growth mechanisms in polluted environments? How does this study differ from these previous studies?

Response:

We have reorganized the introduction section as suggested.

Firstly, we emphasize that the importance of NPF in the polluted regions, where it plays a crucial role in contributing to atmospheric particle number concentrations and driving haze formation. Nevertheless, the precise NPF mechanism in the heavily polluted areas remain incompletely understood. Some studies argue that sulfuric acid and various basic molecules are key drivers of NPF, while others suggest that oxygenated organic molecules from anthropogenic sources dominate the NPF process. More references and discussions related to NPF in the polluted areas have been added as suggested.

Secondly, we introduced the Po Valley region in Southern Europe, which is characterized as a typical polluted area where both haze and NPF events occurred, but which is not a megacity environment. Po-Valley is unique due to its geographical location as it is surrounded by mountains from three directions. Moreover, its atmospheric pollution cocktail is induced through a mixture of industrial activities, agriculture (livestock) and to a lesser extent compared to megacities through traffic and heating. The high formation rate (87 $cm^{-3}$ $s^{-1}$) and the presence of a substantial number of sub-3 nm particles indicate that NPF contributes significantly to the particle number concentrations and likely to haze formation in the Po Valley.

Lastly, we emphasized that the studies elucidating NPF mechanism in the polluted areas were rather scarce. In Po Valley, NPF mechanism has not been elucidated due to a lack of direct measurement using the state-of-the-art instrumentations for chemical components of clusters and size distributions of particles from < 3 nm. Therefore, gaining a better understanding of NPF in the Po Valley is essential for effective air pollution control and the implementation of measures to address $PM_{2.5}$ pollution.

We made the following revision to the introduction section:

Line 53-78:

"NPF is closely linked to atmospheric air pollution. Efficient nucleation and growth are crucial factors contributing to haze formation, according for over 65% of the particle number concentrations in urban environment (Kulmala et al., 2021; Guo et al., 2014). Frequent NPF events have also been observed in heavily polluted urban environments, including megacities in China (Chu et al., 2019; Yao et al., 2018; Du et al., 2022) and India (Sebastian et al., 2022). Strong and frequent NPF events have been reported in the most urbanization areas in China, such as the North China Plain (Wang et al., 2015; Wang et al., 2013; Wu et al., 2011; Wu et al., 2007; Shen et al., 2011), Yangtze River Delta (Dai et al., 2017; Yu et al., 2016; Xiao et al., 2015) and Peal River Delta (Yue et al., 2013; Peng et al., 2014; Liu et al., 2008). This observation contradicts theoretical calculations that suggest NPF events are less likely to occur in polluted areas, where high levels of preexisting aerosols acting as condensational sinks (CS) are capable of quickly scavenging gaseous precursors of NPF (Kulmala et al., 2017).

The elucidation of NPF precursors and mechanisms has varied among different sampling locations and studies. No uniform theory or mechanism can elucidate the NPF occurrence in different polluted areas or in different seasons. For example, in Shanghai and Beijing, China, sulfuric acid (SA, $H_2SO_4$) and amines were identified as key contributors to initial particle formations (Yao et al., 2018; Cai et al., 2021; Yan et al., 2021). On the other hand, some studies also suggests that photooxidation products of vehicle emitted organic vapors, dominate NPF in urban conditions rather than SA or base species (Guo et al., 2020). Meanwhile, in Barcelona, Spain, which is significantly less polluted than Asian megacities but still shows frequent high pollution levels, NPF was reported to be associated with SA along with highly oxygenated organic molecules (HOMs) (Brean et al., 2020). The discrepancies in the reported NPF mechanisms may arise from the limited utilization of state-of-the-art instruments, such as those capable of measuring size distribution down to 1-2 nm and directly identifying clusters and vapors with the influences by spatio-temporal variations (Wang et al., 2017). Therefore, gaining a better knowledge of the key participants, nucleation mechanism and the roles of pre-existing particles is crucial for comprehending the causes of the high NPF frequencies in polluted regions. This knowledge can be essential for developing effective local $PM_{2.5}$ control and implementation strategies."

Line 83-88:

"…make the Po Valley region a hotspot in Europe for high aerosol loadings (Saarikoski et al., 2012; Li et al., 2014; Finzi and Tebaldi, 1982; Daellenbach et al., 2023). But it is distinct from Asian megacities as the population density is significantly lower (250 people $km^{-2}$ in Po Valley compared to e.g., 1 400 people $km^{-2}$ in Beijing), resulting in effects such as traffic or residential heating being less dominant pollution sources."

And Line 96-102

"While previous studies conducted in the Po Valley have reported frequent NPF events characterized by high nucleation and growth rates, the clustering mechanism and the dominant precursors for particle growth have not been investigated to-date. Especially with respect to the distinct features of Po-Valley compared to the more intensely researched megacity environments, a deeper understanding of frequent NPF events, including their precursors, nucleation mechanisms, and growth processes is crucial for air pollution control and the effective implementation of $PM_{2.5}$ mitigation measures in such semi-urban but highly industrialized regions as Po Valley. In this study, we conducted a 2-month field campaign in the months of March – April 2022…"

4. In Section 2.3.2, Condensation sink, nucleation and growth rate calculations should give the calculation formula.

Response:

The condensation sink, nucleation and growth rate calculations and equations have been added as follows in the method section.

Line 236-240

"The growth rates were calculated by first determining the time to reach 50% of the maximum concentration and then the average growth rate is derived as the slope of the linear fit between the time and diameter:

$$GR = \frac{\Delta d_p}{\Delta t} \approx \frac{d_{p,f} - d_{p,i}}{t_f - t_i},$$

where $d_{p,f}$ is the diameter at the end time $t_f$ and $d_{p,i}$ is the diameter at the start time $t_i$."

Line 245-247

"

Formation rates were then calculated by rearranging the equation describing the time evolution of the particle size distribution. Formation rate for a given diameter $d_{p1}$ is calculated as

$$J_{dp1} = \frac{dN_{dp1-dp2}}{dt} + CoagS_{dp1} \cdot N_{dp1-dp2} + \frac{GR}{\Delta d_p} N_{dp1-dp2},$$

"

5. Sections 3.1 and 3.2 are merged.

Response:

Thanks for the reviewer's comment. We have made the following changes to sections 3.1 and 3.2. We have relocated the discussion on formation rate from section 3.1 to 3.2, thereby separating the discussions of new particle formation frequency and nucleation mechanism.

In the revised manuscript, in section 3.1, we discussed the NPF frequency, formation rates and sulfuric acid concentrations during our measurement periods. We further compare these results with previous studies conducted in the Po Valley region. In section 3.2, our focus is solely on the NPF mechanism in the Po Valley. We achieve this by comparing our finds with the CLOUD chamber experiments under various basic molecule levels, using kinetic models, and examining the predicted growth rates with different condensable vapors (i.e., sulfuric acids and OOMs).

We have integrated the following revised content from the original section 3.2 into 3.1:

Line 298 to 307:

"SA has long been known as a primary gaseous precursor for NPF in continental environments, owing to its extremely low volatility (Kirkby et al., 2011; Kulmala et al., 2013). During our sampling period, we observed high SA concentration in the Po Valley, in accordance with the frequent NPF events. The daily average SA concentration measured between 10:00 – 14:00 LT was $4.6\times10^6$ cm$^{-3}$, which increased to $8.5\times10^6$ cm$^{-3}$ during NPF events with growth, aligning with previous findings from the same site ($1.6\times10^7$ cm$^{-3}$ during NPF in summer of 2009, (Paasonen et al., 2010)). Over the entire sampling period (10:00 – 14:00 LT), SA showed a moderately correlation with the calculated $J_{1.7}$ ($r = 0.49$, Spearman correlation coefficient, for the logarithmic values), but its relationship varied among different days. This suggests that in addition to SA, other components, such as basic molecules, may also contribute to driving NPF events and subsequent growth in the Po Valley."

6. Section 3.3: It is necessary to add some in-depth analysis of mechanisms. For example, what are the commonalities compared with research results in Shanghai, Beijing and New Delhi? What are the differences in the Po Valley's meteorological conditions, chemical compositions and aerosol background concentrations compared with these polluted areas? What impact do these differences have on the NPF incident? Is the growth mechanism of new particles in the NPF event in Po Valley similar to that in other polluted environments? What are the effects of being highly polluted?

Response:

Following Comment 7's suggestion, we have relocated the comparison between Shanghai, Beijing, New Delhi and Po Valley to Section 3.4 of the revised manuscript. In Section 3.3, our focus centers on the chemical compositions of ions and neutral clusters directly associated with NPF events in Po Valley itself. We delve into the distinctions in NPF mechanism, meteorological conditions, cluster compositions and background pre-existing particle chemical compositions between the Po Valley and other polluted environments (e.g., Beijing and Shanghai), which are further discussed in Comment 7.

In Section 3.3, we have made the following revisions to better illustrate the disparities between NPF and non-NPF days and to highlight the impact of the polluted environment on the NPF events.

Line 439-447:

"Throughout the entire sampling period, relatively high concentrations of fine particulate matters (PM$_{2.5}$) were measured, with a daily average of 17 µg m$^{-3}$ and a maximum value of 43 µg m$^{-3}$. Correspondingly, the hourly CS levels, which quantify the ability of pre-existing particles to scavenge gaseous precursors, ranged from $<1\times10^{-4}$ s$^{-1}$ to $3\times10^{-2}$ s$^{-1}$ with an average value of $5.4\times10^{-3}$ s$^{-1}$. Previous studies in polluted areas, such as Chinese megacities, have shown that NPF events are closely linked to CS levels (Cai et al., 2017). NPF probability was reported to decreased to 50% when CS was around $1\times10^{-2}$ s$^{-1}$ and completely shut off with CS of $6\times10^{-2}$ s$^{-1}$ (Du et al., 2022). However, in the Po Valley, we observed no strong influence of CS on NPF events, with only a slightly difference in CS during the noontime of non-NPF days (median: $9.4\times10^{-3}$ s$^{-1}$) than NPF days (median: $8.6\times10^{-3}$ s$^{-1}$)."

7. The discussion in Section 3.4 is too simple and requires in-depth analysis to compare the similarities and differences in the generation and growth mechanisms under different atmospheric environments. In particular, the unique results of Po Valley need to be highlighted instead of simply comparing the differences in data values as currently done.

Response:

We thank the reviewer for this constructive suggestion. We have incorporated the following discussions related to the comparison between the NPF mechanism in the Po Valley and other polluted regions, as suggested:

To compare the chemical compositions of the pre-existing particles between the Po Valley and other polluted regions, we have included the following discussions:

Line 469-486

"NPF events in Po Valley are not strongly dependent on the CS levels ($9.4\times10^{-3}$ $s^{-1}$ and $8.6\times10^{-3}$ $s^{-1}$ for non-NPF and NPF days, respectively), likely due to generally lower CS levels than the Asian megacities (Fig. S8). The strength of precursor sources and their accumulation in the Po-Valley region might thus be more important for NPF to occur than the overall pre-existing sink for those precursors.

The average $PM_1$ concentrations during the sampling period was around 8 µg $m^{-3}$, significantly lower than New Delhi (268 µg $m^{-3}$), Beijing (33 µg $m^{-3}$, (Li et al., 2019)) and Shanghai (30 µg $m^{-3}$, (Song et al., 2023), Fig. S9). The major chemical compositions in $PM_1$ in Po Valley were similar to those in Beijing and Shanghai, with organics, ammonium nitrate, and ammonium sulfate being the most abundant components. However, $PM_1$ compositions in New Delhi differed from Po Valley and megacities in China. In New Delhi, strong biomass burning emissions with a high abundance of primary organics (155 µg $m^{-3}$, 58%) suppressed NPF events during the daytime from January to February but led to nocturnal particle growth, which is not observed in other polluted areas (Mishra et al., 2023).

Even with similar levels of CS and total $PM_1$ concentrations (NPF: 6.3 µg $m^{-3}$ and non-NPF: 6.5 µg $m^{-3}$) observed during noontime in Po Valley, the concentration of $NO_3^-$ increased by 50% on non-NPF days compared to NPF days, higher than the increase of $PM_1$ (3.1%) as shown in Fig. S9. A lower CS efficiency on NPF days due to lower fraction of nitrate was reported to suppress the scavenge of NPF precursors in Beijing (Du et al., 2022), which may also have the similar influence in the Po Valley."

[Figure]

Fig. S9. The comparison of $PM_1$ compositions between Po Valley and other polluted cities. $PM_1$ compositions in (a) Po Valley for all days, (b) Po Valley for NPF with growth days, (c) Po Valley for non-NPF days, (d) Beijing, (e) Shanghai, (f) New Delhi. The green, blue, red, orange, chloride and black colors represented organics, nitrate, sulfate, ammonium, chloride, and BC, respectively. The NR-PM1 is from a co-located aerosol mass spectrometer measurement (Paglione et al., 2020) and BC concentration is from the MAAP measurement with $PM_{2.5}$ inlet. The chemical compositions of Beijing (Li et al., 2019), Shanghai (Song et al., 2023), and Delhi (Mishra et al., 2023) are from previous literature.

Line 453-464:

"The average SA concentrations ($4.6\times10^6$ $cm^{-3}$, 10:00 – 14:00 LT) were comparable to the levels observed in polluted megacities in China (ranging from $3.9\times10^6$ $cm^{-3}$ to $7.4\times10^6$ $cm^{-3}$, Fig. 5c), but significantly higher than those in remote areas like Hyytiälä ($9\times10^5$ $cm^{-3}$) and the Jungfraujoch ($5\times10^5$ $cm^{-3}$)... SA concentrations during NPF days ($8.6\times10^6$ $cm^{-3}$) in the Po Valley were twice as high as those on non-NPF days ($4\times10^6$ $cm^{-3}$). This difference may be linked to the significant variations (t-test, $p<0.05$) of $SO_2$ concentrations between NPF days (0.38 ppbv) and non-NPF days (0.20 ppbv). This contrasts

with findings in Beijing, where similar or even higher levels of SA and SO$_2$ were observed during non-NPF days compared to NPF event days (Yan et al., 2021). The variations in SO$_2$ and SA concentrations in the Po Valley could possibly be attributed to differences of air masses, as indicated by higher RH on non-NPF days (53%) than on NPF days (38%) but similar temperature (NPF days: 288 K, non-NPF days: 287 K). On higher RH days, photochemistry may be suppressed, potentially reducing the formation of sulfuric acid and low volatile condensable vapors."

For the comparison of the NPF mechanism, we added:

Line 502-519

"Therefore, in the Po Valley region, the initial nucleation of frequent NPF is primarily attributed to high sulfuric acid concentrations and basic molecules, including ammonia and various amines. This mechanism is generally similar to what is observed in Chinese megacities. However, in the Po Valley region, DMA, a typical base in anthropogenic emission-influenced areas, is insufficient to stabilize the high levels of sulfuric acid, leading to the involvement of other basic molecules like additional other type of amines and ammonia, likely originating from fertilization in the area. This involvement of ammonia and other amines differs from Chinese megacities such as Shanghai, where high levels of DMA were observed (~40 ppt, (Yao et al., 2018; Yao et al., 2016)). As insufficient DMA is available to stabilize all clusters, we speculate that the clustering is therefore sensitive to the abundance of amines and the variations in DMA or other amine concentrations would result in different formation rates. In that sense, during our sampling period, NPF in Po Valley seems to be more sensitive to the strength of certain emission sources of amines compared to megacity environments, where the clustering is "saturated" with respect to DMA (i.e., proceeding at the maximum kinetically possible rate). The abundant OOMs dominate the consecutive growth process, leading to a comparable GR to Chinese megacities such as Beijing and Shanghai. The relatively lower CS in the Po Valley allows the newly formed particles to have a higher survival probability. Therefore, NPF can provides more long-term surviving particles in the Po Valley, highlighting its decisive role for Po-Valley aerosol and PM$_{2.5}$ concentrations."

8. Line 268, "concentrations" should be "concentration." Pay attention to other similar singular and plural expressions in the manuscript.

Response:

Now "concentrations" has been changed to "concentration". The singular/plural expressions throughout the manuscript have been checked and revised as suggested.

9. Line 259, Please add citations for Beijing and Shanghai (59 cm-3 s-1 − 225 cm-3 s-1),

Response:

The reference for the formation rate values have been added.

Now Line 290-292 has been revised as follows:

"The high formation rate, which is comparable with heavily polluted urban environments such as Beijing and Shanghai, China (59 cm$^{-3}$ s$^{-1}$ − 225 cm$^{-3}$ s$^{-1}$ (Deng et al., 2020; Yao et al., 2018))."

10. Line 344, The unit expressions in "GR=6.1 nm h-1" and "(1.0– 2.4) nm/h, 4.6 (2.9 – 5.8) nm/h, and 5.1 (3.8 – 8.8) nm/h" in line 323 should be consistent, which should also be noted in other parts of the article.

Response:

Thanks for reviewer's suggestion. Now the unit expressions have been revised to nm h$^{-1}$ throughout the revised manuscript.

11. Conclusion. The authors should give unique conclusions on the formation and growth mechanism of NPF in the highly polluted environment of Po Valley, especially compared with other polluted environments. Needs to focus on giving quantitative results.

Response:

In order to better present our results, we have revised the conclusions as follows:

Line 521-544

[revised manuscript text omitted]